# Mobile phone carrying locations and risk perception of men: A cross-sectional study

**Berihun M. Zeleke**[1]*, **Christopher Brzozek**[1], **Chhavi Raj Bhatt**[2,3], **Michael J. Abramson**[1,2], **Frederik Freudenstein**[1,2,4], **Rodney J. Croft**[1,4], **Peter M. Wiedemann**[1], **Geza Benke**[1,2]

**1** Centre for Population Health Research on Electromagnetic Energy (PRESEE), School of Public Health and Preventive Medicine, Monash University, Melbourne, Australia, **2** Monash Centre for Occupational and Environmental Health, School of Public Health and Preventive Medicine, Monash University, Melbourne, Australia, **3** School of Clinical Sciences at Monash Health, Melbourne, Australia, **4** Australian Centre for Electromagnetic Bioeffects Research, Illawarra Health and Medical Research Institute, School of Psychology, University of Wollongong, Wollongong, Australia

* berihun.zeleke@monash.edu

**Data Availability Statement:** All relevant data are within the manuscript, supporting information files, and the public repository Figshare. The Figshare data can be obtained at: https://bridges.monash.

## Abstract

Little was known about the relationship between carrying mobile phone handsets by men and their risk perception of radiofrequency-electromagnetic field (RF-EMF) exposure due to carrying handsets close to the body. This study aimed to determine where men usually carried their handsets and to assess the relationship to risk perception of RF-EMF. Participants completed a self-administered questionnaire about mobile phone use, handset carrying locations, and levels of risk perception to RF-EMF. Data were analysed using linear regression models to examine if risk perception differed by mobile phone carrying location. The participants were 356 men, aged 18–72 years. They owned a mobile phone for 2–29 years, with over three quarters (78.7%) having a mobile phone for over 20 years. The most common locations that men kept their handsets when they were 'indoors' were: on a table/desk (54.0%) or in close contact with the body (34.7%). When outside, 54.0% of men kept the handset in the front trouser pocket. While making or receiving calls, 85.0% of men held their mobile phone handset against the head and 15.0% either used earphones or loudspeaker. Men who carried their handset in close contact with the body perceived higher risks from RF-EMF exposure compared to those who kept it away from the body (p<0.01). A substantial proportion of men carried their mobile phone handsets in close proximity to reproductive organs i.e. front pocket of trousers (46.5%). Men who kept their handset with the hand (p < .05), and those who placed it in the T-shirt pocket (p < .05), while the phone was not in use, were more likely to perceive health risks from their behaviour, compared to those who kept it away from the body. However, whether this indicates a causal relationship, remains open.

## Introduction

These days in most countries, almost every adult carries a (smart) mobile phone handset. Mobile phones are increasingly becoming one of the most ubiquitous digital tele-communication tools. According to the International Telecommunication Union, 85% of the global

edu/articles/dataset/Men_data_final_2021_dta/17308361.

**Funding:** This research project is supported by the Centre for Population Health Research on Electromagnetic Energy (PRESEE), School of Public Health and Preventive Medicine, Monash University. The centre is funded by a grant from the National Health and Medical Research Council, Australia (APP 1060205). The funders had no role in study design, data collection and analysis, decision to publish, or preparation of the manuscript.

**Competing interests:** The authors declare no competing interests

population have access to telecommunication networks and 57% to internet services [1]. In addition to making and receiving calls, digital technologies available in smartphones enable access to several services such as voice data (e.g., making or receiving a call), and video data (messenger, WhatsApp, YouTube, etc.) [2]. Mobile phones emit and receive the data via radio-frequency electromagnetic fields (RF-EMF).

Mobile phone usage related RF-EMF exposure and likely human health effects have long been of concern internationally [3, 4]. RF-EMF exposure from mobile phones may induce some heating effect and/or a sensation of warmth, depending upon a range of factors, including the distance between the mobile phone and part of the body exposed during active use [5, 6]. A recent systematic review reported that studies speculated that scrotal overheating might affect male fertility [7]. In another observational study, men who carried their mobile phones in their hip pockets or on their belts had lower sperm motility than men who did not carry a mobile phone or who carried their mobile phone elsewhere on the body [8]. In view of this, understanding of people's habits of placing/carrying a mobile phone becomes important. Though the extent of mobile phone carrying habits have not been adequately assessed, some data showed that people tend to place or carry their mobile phone close to the body [5–7].

Previous studies that assessed people's preferences of carrying mobile phones identified both cultural and gender differences as to where people carry them [9, 10]. For instance, among Australian women aged 15–40 years, almost three quarters (72%) carried the mobile phone in a pocket below waist level, with 57% doing so over the preceding week [10]. However, comparable data amongst men were scarce.

The speculated risk of mobile phones carrying and use in relation to fertility issues remains on the agenda in both mainstream media [11–13] and scientific literature [14–17]. Therefore, given the widespread use of mobile phones by men, it seems reasonable to investigate in more detail how men perceive their risks related to RF-EMF from carrying mobile phones close to body parts.

The general public is fairly aware of different RF-EMF sources and risk associated with RF-EMF exposures to the human body [4]. Studies have shown that non-experts' risk perceptions of RF-EMF sources are influenced by subjective exposure perception [3, 4]. There seem to be some public concerns about the potential health effects of carrying or placing mobile phones close to the body [18]. Within this context, the relationship between mobile phone RF-EMF exposure perception, risk perception, and the actual mobile phones carrying habit (close to the body or not) becomes an important area of research. Hence, the current study aimed to determine: i) where men usually carried their mobile phones, and ii) whether this was related to their perceptions of risk concerning exposure to RF- EMF, primarily emanating from their own mobile phones.

## Methods

### Study design, participant recruitment and data collection

This study was a cross-sectional survey of 356 men who completed a self-administered hard-copy questionnaire. Sample size was estimated by a single population proportion formula with the assumption of 95% confidence level, 5% margin of error, 5% non-response rate, and 50% having carrying locations close to body parts, similar to those reported for women [10]. Participants were recruited to participate in the study via advertisements posted on notice boards at public libraries, universities, and hospitals across Melbourne, Australia. Participants were also individually approached at sporting clubs and invited to participate in the study. Data collection was undertaken between October 2018 and February 2019.

A structured questionnaire that inquired about socio-demographic variables (age, educational level, residential postcode, ethnicity, and occupational description) and mobile phone use and handset carrying location related information, was used in this survey. Mobile handset carrying locations of men were assessed by asking them to rate, on a Likert-scale (1 = never; 2 = rarely, 3 = occasionally, 4 = often, 5 = very often), how commonly they carry it with their hand, keep it on a table, in a bag, or in the pocket of their T-shirt, Jacket or Trousers (front or back pockets).

### Risk perception

Participants were asked to rate their level of personal risk perception to RF-EMF exposure due to their own most-frequent mobile phone carrying behaviour. This was measured using questions also formatted on Likert-scales, using the question "On a scale of 1-to-10 (1 = not-at-all,. . . 10 = very dangerous), how dangerous do you consider the RF-EMF from mobile phones communication devices are in terms of increasing cancer risks to people?" and "On a scale of 1–5 (1 = never, . . ..., 5 = very often), how often do you think that carrying a mobile phone in your most common carrying habit poses a health risk to yourself?".

Prior research by Wiedemann and colleagues [19] showed that the thematic relevance, i.e. how often people think about a risk issue, is a good predictor of people's real-life risk perception, reducing bias in the survey situation as concerns expressed in questionnaires do not always reflect concerns in everyday life. Using a similar Likert scale, we assessed how often people thought carrying their mobile phone posed a health risk to them, which was stated as: "*On a scale of 1-to-5, (1 = never. . ..5 = very often), how often do you think that carrying a mobile phone in your most common carrying habit poses a health risk to yourself*?"

### Ethical considerations

Ethics approval was obtained from Monash University Human Research Ethics Committee (project ID:14080). Participants provided written informed consent.

### Data analysis

Descriptive statistics are presented as frequencies, percentages, means (SD) and ranges depending on the distribution. Using postcodes, participants' residential locations were classified into metropolitan or non-metropolitan areas according to the Australian Bureau of Statistics Geographical classification [20]. Overall, handset carrying locations of men were categorized into: while *"not using the phone"*, during *"passive use"* (e.g. listening to music/ radio/podcast online with the phone), and while making or receiving calls *("active use")*. Furthermore, mobile phone locations while making/receiving calls were classified into three main categories as either *"against the ear with a hand"*, *"using earphones"*, or *"using speaker"*. Handset carrying locations were rated based on frequency (never, rarely, occasionally, often, very often). For analysis purposes, these were categorized into binary variable taking often/very often, as a positive response.

Personal risk perception levels were calculated and compared by handset carrying locations and demographic variables. Data were analysed to examine if the risk perception of men differed significantly between occupational groups, and also by carrying locations. Between group differences were assessed using independent t-tests. Associations between outcome and predictor variables were investigated using linear regression models. Factors identified to be associated with outcome variables ($p<0.2$) were further investigated by using simultaneous multiple regression models. All analyses were performed using STATA version 15.0 (Stata-Corp, College Station, TX) and p-values $<0.05$ considered statistically significant.

## Results

### Characteristics of study participants

Participants were aged between 18 and 72 years. One participant who did not own a mobile phone was excluded from further analysis. Most participants (89.0%) were recruited from metropolitan areas of Melbourne, and over two-thirds (69.8%) were educated beyond high school. Nearly half of the participants (47.7%) were Caucasian; while 37.5% identified themselves as of Asian origin (Table 1).

**Table 1. Participants' socio-demographic characteristics.**

| Variables | Frequency (n) | Percentage (%) |
|---|---|---|
| **Age, mean±SD (years)** | 33.3±11.1 | |
| 18–24 | 72 | 20.2 |
| 25–34 | 154 | 43.3 |
| 35–44 | 73 | 20.5 |
| 45+ | 57 | 16.0 |
| **Ethnicity** | | |
| Caucasian | 167 | 47.4 |
| Asian | 132 | 37.5 |
| Other | 53 | 15.1 |
| **Residential location** | | |
| Metropolitan | 317 | 89.0 |
| Non-metropolitan | 5 | 1.4 |
| Missing | 34 | 9.6 |
| **Education** | | |
| High school or less | 57 | 16.1 |
| Vocational training (diploma) | 50 | 14.1 |
| Bachelor's degree | 126 | 35.6 |
| Postgraduate degree | 121 | 34.2 |
| **Occupation** | | |
| Administrative and Finance | 53 | 15.0 |
| Healthcare professional (medical or nursing) | 46 | 13.0 |
| Service Sector | 78 | 22.1 |
| Education and/or research | 69 | 19.6 |
| Student | 70 | 19.8 |
| Other | 37 | 10.5 |
| **Years since first owned a mobile phone (mean±SD)** | 14.7±5.2 | (range 2–29) |
| **Mobile phone mode during daytime** | | |
| Standby | 339 | 96.3 |
| Flight mode or turned off | 13 | 3.7 |
| **Mobile phone mode during sleep** | | |
| Standby | 287 | 82.0 |
| Flight mode | 26 | 7.4 |
| Switched off | 87 | 10.6 |
| **Laterality while talking on the mobile phone** | | |
| Right | 140 | 56.7 |
| Left | 59 | 23.9 |
| Almost equally | 48 | 19.4 |

## Mobile phone use and carrying locations

The participants owned a mobile phone for an average of 14.7 (SD 5.2) years, while over three quarters (78.7%) owned a mobile phone for over 20 years.

The handset carrying locations of men are presented in Fig 1. The most common locations that men kept their mobile phones when they are 'indoors' (at home or work), and the phone 'not in use', were on a table/desk (54.0%) and in close contact with the body (34.7% placed it either in pockets or held by hand). Of the men who keep their mobile phone in close contact to their body parts, 46.0% keep it in the trouser front pocket. When outside, and not using their mobile phone, 54.0% of men keep their mobile phones in the trouser front pocket, and 46.0% in a bag or in a vehicle or somewhere other than pockets.

While using the mobile phone passively (e.g. listening to music with an earphone plugged into the handset), 50% placed it in the trouser front pocket. While sleeping, 82% of men kept their mobile phone on "*standby*", 7.4% in "*flight-mode*" and 10.4% "*switched-off*".

While making or receiving calls, most (85%) of the men held the handset against the head, of whom over half (56.7%) held it against the right ear, 23.9% with the left ear, and 19.4% against either ear almost equally (Fig 2). The rest (15%) either used earphones or a loudspeaker while making/receiving calls with their mobile phones.

## Mobile phone use, risk perception and carrying location

Over a third (36.7%) of men rated their RF-EMF exposure as high or very high, and 21.5% considered it "dangerous" or "very dangerous" to talk using the mobile phone against the ear.

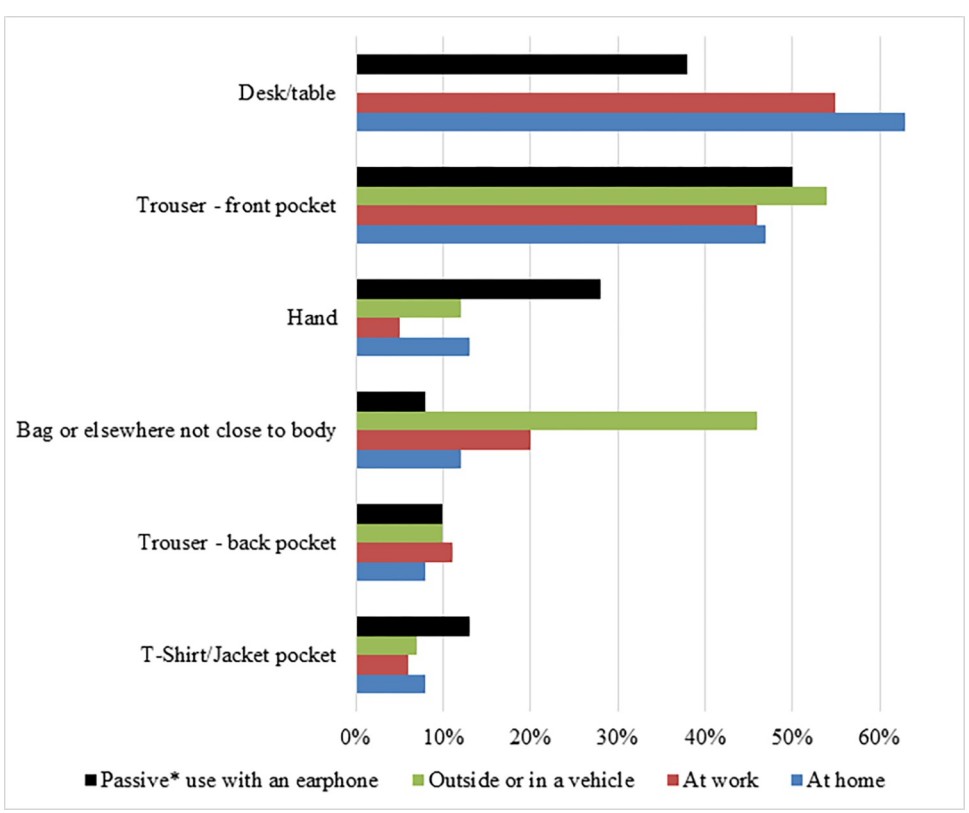

**Fig 1. Mobile phone locations when not in use or only passive use** (*passive use: E.g. listening to music/radio/ podcast online).

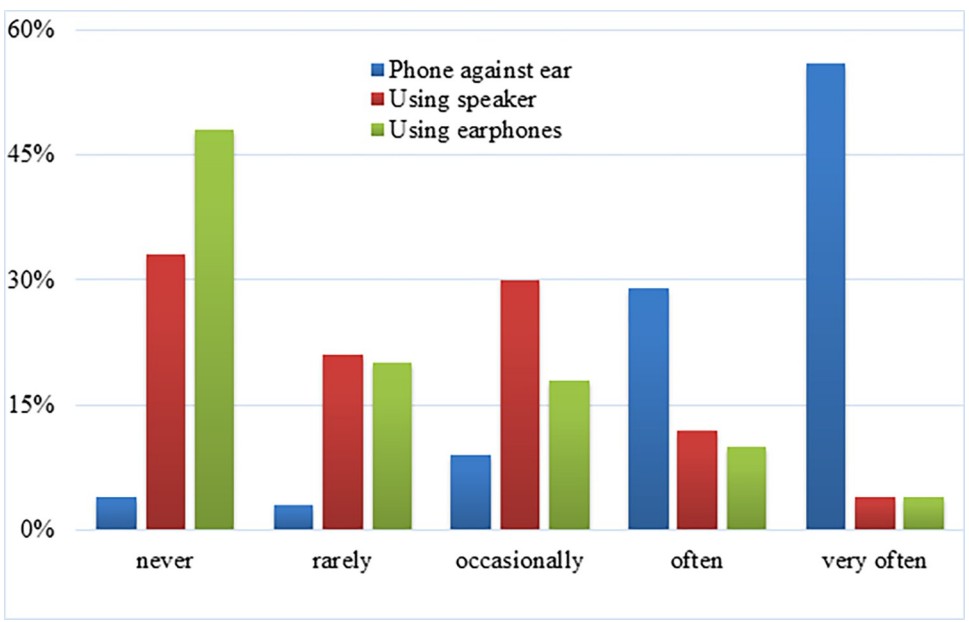

**Fig 2. Mobile phone carrying locations of men while making/receiving calls (active use).**

Likewise, 14.1% of men perceived a high or very high risk of health issues related to their usual handset carrying location, and 13.2% considered changing the location in the future. Overall, 29% of men reported feeling a heating effect often or very often on body parts near where the mobile phone was carried, and 16.0% believed that carrying mobile phones could cause infertility.

Risk perception due to personal mobile phone carrying locations was higher amongst men who carried their mobile phone in close contact with their body, compared to those who carry it away from the body (Mean values for risk perception: 2.54 vs. 2.20; β = 0.336, p<0.01). For instance, about a third (31.6%) of men who often/very often kept their handset in the trouser pocket rated their risk to be high or very high, compared to 13.7% of those who did so less frequently. The proportion of those who reported a high/very high level of risk perception in relation to their phone carrying habits is presented in Fig 3. Among men who carry their phone in the trouser pocket, 33% reported a high/very high level of risk perception.

Table 2 presents linear regression models for the association between risk perception due to the mobile phone carrying location and predictor variables. After adjusting for potential confounders, the findings indicated that students (β = -0.650, p = 0.003) and participants who identified themselves as Caucasian (β = -0.554, p < .001) have lower risk perception levels. Men who carried their mobile phones with the hand when not using it (β = 0.377, p < .05), and those who placed it in the shirt pocket (β = 0.608, p < .05) were more likely to report higher risk perception levels, compared to those who placed it in a bag or on a table (Table 2). In addition, men who perceived risk due to their mobile phone carrying location were three times more likely (OR = 3.01; 95%CI 1.18–7.70) to report heat around the body part where the mobile phone was usually carried.

## Discussion

This study documented the most frequent locations men carried their mobile phones, when not in use, and carried off-body. Half the men, often or very often, carried their mobile phone

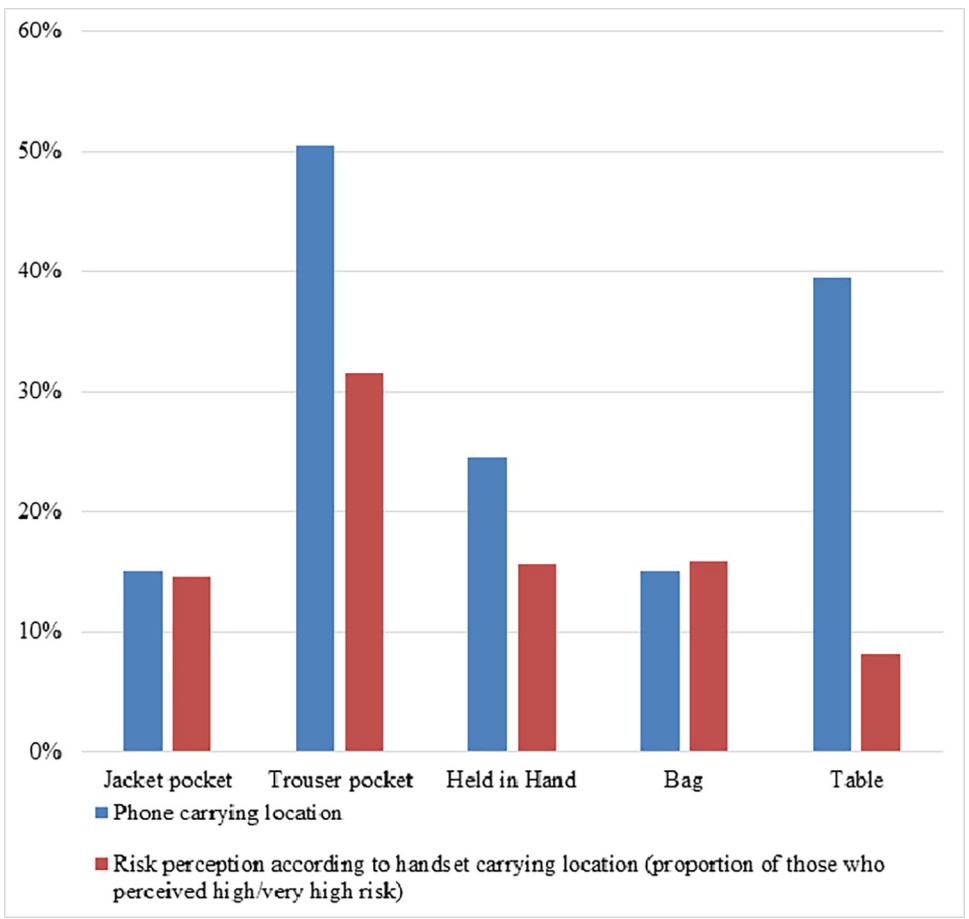

**Fig 3. Mobile phone carrying locations and risk perception of men.**

handset in the trouser pocket. Men who carried their handset in close contact with their body perceived higher risks from overall RF-EMF exposure, compared to those who kept it away from the body. Men who carried their handset in close contact with their body perceived higher risks from RF-EMF exposure and their mobile phone handset carrying behaviour.

These findings are comparable to those reported by previous studies elsewhere [9, 10, 16, 17]. For example, up to 60% of men (compared to 16.4% of women) carried their mobile phones in their trouser pockets [9]. Similarly, higher proportion of women (61.1% compared to 10.1% of men) tended to carry their mobile phones in their hand bags [9, 10, 16, 17]. In Australia, the most common locations where women placed their mobile phones were off-the body (86%), in the hand (58%), a skirt/trouser pocket (57%), or against the breast (15%) when the mobile phone is not in use or during passive use [10].

The results of the current study would translate to a pattern of the whole body and reproductive organ specific RF-EMF exposures from mobile phone use and carrying behaviour, and resulting exposure and risk perceptions. There is some evidence that perception of risk may impact behaviour to some extent such as whether or not to carry the handset close to reproductive organs [8]. Men who kept their mobile phone away from body-parts while it was not in use were half as likely to perceive health risks from the specific behaviour. If one carries his mobile phone close to body parts, he might be prompted by being inquired about possible negative effects and may rate higher levels of risk perception.

**Table 2. Linear regression models for selected predictor variables and risk perception to own mobile phone carrying location.**

| | Risk perception to own mobile phone handset carrying location | | | |
|---|---|---|---|---|
| | Univariate analysis | | Multivariable analysis | |
| | β | *p*-value | β | *p*-value |
| **Age (years)** | 0.003 | 0.632 | 0.006 | 0.751 |
| **Educational status** | | | | |
| Beyond high school vs. High school or less | 0.110 | 0.397 | -0.079 | 0.582 |
| **Ethnicity** | | | | |
| Caucasian vs. non-Caucasian | **-0.504** | **<0.001** | **-0.554** | **<0.001** |
| **Occupation** | | | | |
| Admin & Finance | Ref | | Ref | |
| Healthcare worker | -0.3 | 0.102 | -0.392 | 0.086 |
| Service sector | -0.108 | 0.592 | -0.073 | 0.720 |
| Education/Researcher | -0.235 | 0.254 | -0.219 | 0.286 |
| Student | **-0.608** | **0.003** | **-0.650** | **0.003** |
| Other | -0.446 | 0.063 | -0.387 | 0.114 |
| **Mobile phone carrying location when not in use** | | | | |
| Bag/Table | Ref | | Ref | |
| Trouser pocket | 0.170 | 0.221 | 0.180 | 0.192 |
| T-Shirt (Jacket) pocket | 0.547 | 0.103 | **0.608** | **0.046** |
| Held in hand | 0.355 | 0.053 | **0.377** | **0.041** |

In the current study, one in every six men believed that carrying mobile phones would cause infertility. This may not be surprising since a third of the participants also reported feeling heating on body parts near where the mobile phone was carried. This may be worrisome in modern days since smartphones are more prone to heating human tissues in contact as there is a trend in moving toward lower specific absorption rate (SAR) and higher battery capacity smartphones due to their increased uses other than making/receiving calls [6].

Students and those who identified themselves as Caucasian were less likely to consider their mobile phone carrying locations posed any health risks. In a study of women, younger age groups were found to be best informed about RF-EMF emissions from smartphones and other wireless devices [10]. Being well-informed and educated is known to be an important determinant of trust [21], and hence more educated people may acquire information, build trust, and are likely to report lower levels of risk perception. Men who carried their mobile phones in their T-shirt pockets and those who hand-held them, when it is not in use, were found to have higher levels of risk perception, compared to those who place their phone away from the body. Understanding the most common locations where men carry their mobile phones and whether their risk perception towards RF-EMF exposure and health effects depends on their phone carrying behaviour could help tailor prevention and future public health interventions.

This study, which employed a sample of men recruited from the general public, could form a baseline for future research. Furthermore, data presented in this study could help generate hypotheses regarding the effects of carrying mobile phones close to body parts on reproductive health outcomes in relation to the RF-EMF exposure. Given the ubiquity of mobile phone use and that the majority of men carried their mobile phones in close proximity to reproductive organs (e.g. testes), the relationship between resulting perceived RF-EMF exposures and associated health risks warrants further research. Similarly, reported local heating associated with carrying the mobile phone close to the body, especially reproductive organs, and the subsequent perceived physiological and psychological changes requires further investigation in the

context of different phone carrying locations, preferably from larger and representative samples of men.

A limitation of the study was that all participants were men from the Greater Melbourne region, making our sample not representative of the entire Australian male population. Participants were recruited via advertisements and may be pre-selected based on their access to the invitations and voluntary expression of their interest for participation and hence may not be representative of the source population. In addition, the data were collected in the Australian summer season, the findings could be different if survey was conducted during other seasons (e.g., winter) as men may more often wear jackets or coats providing more options to keep their mobile phones elsewhere.

## Conclusion

A substantial proportion of men carried their mobile phones in their front pockets of trousers, which is in close proximity to reproductive organs. Men perceived higher levels of health risks associated with RF-EMF exposures from mobile phones. Those who kept their handset away from body-parts, while the phone was not in use, were less likely to perceive health risks from their mobile phone carrying behaviours.

## Author Contributions

**Conceptualization:** Berihun M. Zeleke, Christopher Brzozek, Michael J. Abramson, Geza Benke.

**Data curation:** Berihun M. Zeleke.

**Formal analysis:** Berihun M. Zeleke.

**Funding acquisition:** Michael J. Abramson, Rodney J. Croft, Geza Benke.

**Investigation:** Chhavi Raj Bhatt, Frederik Freudenstein.

**Methodology:** Berihun M. Zeleke, Christopher Brzozek, Chhavi Raj Bhatt, Michael J. Abramson, Frederik Freudenstein, Rodney J. Croft, Peter M. Wiedemann, Geza Benke.

**Project administration:** Berihun M. Zeleke, Geza Benke.

**Resources:** Christopher Brzozek, Chhavi Raj Bhatt, Frederik Freudenstein, Peter M. Wiedemann.

**Supervision:** Michael J. Abramson, Geza Benke.

**Visualization:** Chhavi Raj Bhatt, Rodney J. Croft, Peter M. Wiedemann.

**Writing – original draft:** Berihun M. Zeleke.

**Writing – review & editing:** Berihun M. Zeleke, Christopher Brzozek, Chhavi Raj Bhatt, Michael J. Abramson, Frederik Freudenstein, Rodney J. Croft, Peter M. Wiedemann, Geza Benke.

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
