## [Decision Letter · Decision Letter 0]

16 Sep 2021

PONE-D-21-18441Mobile phone carrying locations and risk perception of men: a cross-sectional studyPLOS ONE

Dear Dr. ZELEKE,

Thank you for submitting your manuscript to PLOS ONE. After careful consideration, we feel that it has merit but does not fully meet PLOS ONE’s publication criteria as it currently stands. Therefore, we invite you to submit a revised version of the manuscript that addresses the points raised during the review process.

We look forward to receiving your revised manuscript.

Kind regards,

Sofia Scataglini

Academic Editor

PLOS ONE

Journal Requirements:

“This research project is supported by the Centre for Population Health Research on Electromagnetic Energy (PRESEE), School of Public Health and Preventive Medicine, Monash University. The centre is funded by a grant from the National Health and Medical Research Council, Australia (APP 1060205).”

“This research project is supported by the Centre for Population Health Research on Electromagnetic Energy (PRESEE), School of Public Health and Preventive Medicine, Monash University. The centre is funded by a grant from the National Health and Medical Research Council, Australia (APP 1060205).

4. Please ensure that you refer to Figure 3 in your text as, if accepted, production will need this reference to link the reader to the figure.

Additional Editor Comments (if provided):

Dear authors, Thank you for submitting your work. We have received the reviewers' comments and opinions on your amended paper.

I regret to notify you that I would like to re-consider your submission following an extensive significant rewrite based on reviewers' input.

Reviewers' comments:

Reviewer's Responses to Questions

**Comments to the Author**

1. Is the manuscript technically sound, and do the data support the conclusions?

Reviewer #1: Yes

Reviewer #2: Yes

2. Has the statistical analysis been performed appropriately and rigorously? 

Reviewer #1: Yes

Reviewer #2: Yes

3. Have the authors made all data underlying the findings in their manuscript fully available?

Reviewer #1: No

Reviewer #2: Yes

4. Is the manuscript presented in an intelligible fashion and written in standard English?

Reviewer #1: Yes

Reviewer #2: Yes

5. Review Comments to the Author

Reviewer #1: Summary:

In this article, the authors explore the relationship between mobile phone carrying and the risk perception of RF-EMF exposure due to carrying close to the body. The study is done by asking 356 participants (men) a self questionnaire for their mobile phone use. The authors describe statistics of overall usage when they use mobile phones indoor, outdoor, and when receiving calls. Finally, the authors analyze the survey data with the risk perception. The analysis is done mainly using correlation and linear regression models. They found that men generally carry their phone close to reproductive organs (trouser front pocket). And men who kept their phones away from their body were less likely to perceive health risks from their behaviour.

Overall feedback:

Overall, the dataset and descriptive statistics are well described in the article. However, the introduction part can be improved where authors can give solid statements of why readers should care about the survey. Moreover, there is still a lack of discussion in the article. The authors can improve and solidify the overall message of the paper.

Comments:

Introduction. The authors describe overall research in the introduction well but it is still hard to make a clear statement of why there is a need to analyze the survey from the reader's perspective. There is still a lack of a gap in the first paragraph. It would be better if authors can provide a more solid paragraph of why risk perception is needed in this study. I think it would be good for readers to have a one paragraph about health risks of carrying mobile phones, why risk perception is needed, and why surveys are needed.

Methods. The authors describe the dataset well. I do not see the statement of data availability. It would be good to make an anonymous survey available.

Results and discussions. The author explains why risk perception for men who carry phones close contact is higher. It would be good to explain the result more why they might have higher risk perception. In the discussion, it feels more like describing the data but not much insight with the collected data. Overall, I think the writing can be much stronger if authors provide some more evidence or further discussion. For example, why men who carry phones near their body are more likely to perceive more health risks? And here, if they perceive more risk, why do they still carry their phone near the pocket?

Reviewer #2: In order to publish a research article in a journal, a manuscript should describe a technically sound piece of scientific research with data that supports the conclusions. Experiments must have been conducted rigorously, with appropriate controls, replication, and sample sizes. From this point of view, the article appears to be a survey study instead of a technically sound piece of scientific research. I believe that the article should not be classified as a research article.

6. PLOS authors have the option to publish the peer review history of their article (what does this mean?). If published, this will include your full peer review and any attached files.

Reviewer #1: No

Reviewer #2: No

---

## [Author Response · Author response to Decision Letter 0]

21 Dec 2021

Journal Requirements:

AR: The current version is formatted according to the formatting requirements 

“This research project is supported by the Centre for Population Health Research on Electromagnetic Energy (PRESEE), School of Public Health and Preventive Medicine, Monash University. The centre is funded by a grant from the National Health and Medical Research Council, Australia (APP 1060205).”

“This research project is supported by the Centre for Population Health Research on Electromagnetic Energy (PRESEE), School of Public Health and Preventive Medicine, Monash University. The centre is funded by a grant from the National Health and Medical Research Council, Australia (APP 1060205).

AR: We have now deleted the funding in the acknowledgement section of the main manuscript. Please insert the following information regarding funding on our behalf. “This research project was supported by the Centre for Population Health Research on Electromagnetic Energy (PRESEE), School of Public Health and Preventive Medicine, Monash University. The centre was funded by a grant from the National Health and Medical Research Council, Australia (APP 1060205). The funder had no role in the study design, data collection and analysis, decision to publish, or preparation of the manuscript.”

AR: Thank you. We have now removed the phrase as the data related to these were not a core part of our study. 

4. Please ensure that you refer to Figure 3 in your text as, if accepted, production will need this reference to link the reader to the figure.

 AR: Figure 3 is now cited in text

Reviewers' comments:

Reviewer's Responses to Questions

Comments to the Author

1. Is the manuscript technically sound, and do the data support the conclusions?

Reviewer #1: Yes

Reviewer #2: Yes

2. Has the statistical analysis been performed appropriately and rigorously?

Reviewer #1: Yes

Reviewer #2: Yes

3. Have the authors made all data underlying the findings in their manuscript fully available?

Reviewer #1: No

Reviewer #2: Yes

AR: Thank you. We have now removed the phrase as the data related to these were not a core part of our study. 

4. Is the manuscript presented in an intelligible fashion and written in standard English?

Reviewer #1: Yes

Reviewer #2: Yes

5. Review Comments to the Author

Reviewer #1: Summary:

In this article, the authors explore the relationship between mobile phone carrying and the risk perception of RF-EMF exposure due to carrying close to the body. The study is done by asking 356 participants (men) a self questionnaire for their mobile phone use. The authors describe statistics of overall usage when they use mobile phones indoor, outdoor, and when receiving calls. Finally, the authors analyze the survey data with the risk perception. The analysis is done mainly using correlation and linear regression models. They found that men generally carry their phone close to reproductive organs (trouser front pocket). And men who kept their phones away from their body were less likely to perceive health risks from their behaviour.

Overall feedback:

Overall, the dataset and descriptive statistics are well described in the article. However, the introduction part can be improved where authors can give solid statements of why readers should care about the survey. Moreover, there is still a lack of discussion in the article. The authors can improve and solidify the overall message of the paper.

AR: Thank you very much. In the current version of the manuscript, we have revised the introduction and the changes we made are reflected in the tracked changes 

Comments:

Introduction. The authors describe overall research in the introduction well but it is still hard to make a clear statement of why there is a need to analyze the survey from the reader's perspective. There is still a lack of a gap in the first paragraph. It would be better if authors can provide a more solid paragraph of why risk perception is needed in this study. I think it would be good for readers to have a one paragraph about health risks of carrying mobile phones, why risk perception is needed, and why surveys are needed.

AR: We have now revised paragraph 1 and paragraph 2 of the introduction and we have added the following, “ According to the International Telecommunication Union, 85% of the global population have access to telecommunication networks and 57% to internet services (1). In addition to making and receiving calls, digital technologies available in smartphones enabled access to several services such as voice data (e.g., making or receiving a call), and video data (messenger, WhatsApp, YouTube, etc.) (2). Mobile phones emit and receive the data via radiofrequency electromagnetic fields (RF-EMF). 

Mobile phone usage related RF-EMF exposure and likely human health effects have long been of concern internationally (3, 4). RF-EMF exposure from mobile phones may induce some heating effect and/or a sensation of warmth, depending upon a range of factors, including the distance between the mobile phone and part of the body exposed during active use (5, 6). A recent systematic review reported that studies speculated that scrotal overheating might affect male fertility (7). In another observational study, men who carried their mobile phones in their hip pockets or on their belts had lower sperm motility than men who did not carry a mobile phone or who carried their mobile phone elsewhere on the body (8). In view of this, understanding of people’s habits of placing/carrying a mobile phone becomes important.”

Methods. The authors describe the dataset well. I do not see the statement of data availability. It would be good to make an anonymous survey available.

AR: Thank you. Data would be made available once other upcoming analysis is completed

Results and discussions. The author explains why risk perception for men who carry phones close contact is higher. It would be good to explain the result more why they might have higher risk perception. In the discussion, it feels more like describing the data but not much insight with the collected data. Overall, I think the writing can be much stronger if authors provide some more evidence or further discussion. For example, why men who carry phones near their body are more likely to perceive more health risks? And here, if they perceive more risk, why do they still carry their phone near the pocket?

AR: This was a cross-sectional survey with limitations inherent to the design. It was not possible to tell which came first –exposure or outcome? In the case of carrying mobile phone close to body parts and risk perception, we suspect that if one carried his mobile phone close to body parts, he might be prompted by being inquired about it and hence reminded of negative perceived effects. Furthermore, risk perception may not be translated to action such as not carrying mobile phones close to body parts. We have revised the discussion section and it now reads as, “If one carries his mobile phone close to body parts, he might be prompted by being inquired about possible negative effects and may rate higher levels of risk perception.”

Reviewer #2: 

In order to publish a research article in a journal, a manuscript should describe a technically sound piece of scientific research with data that supports the conclusions. Experiments must have been conducted rigorously, with appropriate controls, replication, and sample sizes. From this point of view, the article appears to be a survey study instead of a technically sound piece of scientific research. I believe that the article should not be classified as a research article.

AR: We respectfully disagree with this comment. Although we agree that experimental studies are considered gold standard designs, cross-sectional studies are also valid study designs widely employed in Epidemiology, clinical, social and public health research. 

---

## [Decision Letter · Decision Letter 1]

12 Apr 2022

PONE-D-21-18441R1Mobile phone carrying locations and risk perception of men: a cross-sectional studyPLOS ONE

Dear Dr. ZELEKE,

Thank you for submitting your manuscript to PLOS ONE. After careful consideration, we feel that it has merit but does not fully meet PLOS ONE’s publication criteria as it currently stands. Therefore, we invite you to submit a revised version of the manuscript that addresses the points raised during the review process.

We look forward to receiving your revised manuscript.

Kind regards,

Prof. Anat Gesser-Edelsburg, Ph.D.

Academic Editor

PLOS ONE

Reviewers' comments:

Reviewer's Responses to Questions

**Comments to the Author**

1. If the authors have adequately addressed your comments raised in a previous round of review and you feel that this manuscript is now acceptable for publication, you may indicate that here to bypass the “Comments to the Author” section, enter your conflict of interest statement in the “Confidential to Editor” section, and submit your "Accept" recommendation.

Reviewer #3: (No Response)

Reviewer #4: All comments have been addressed

2. Is the manuscript technically sound, and do the data support the conclusions?

Reviewer #3: Yes

Reviewer #4: Yes

3. Has the statistical analysis been performed appropriately and rigorously? 

Reviewer #3: Yes

Reviewer #4: Yes

4. Have the authors made all data underlying the findings in their manuscript fully available?

Reviewer #3: Yes

Reviewer #4: Yes

5. Is the manuscript presented in an intelligible fashion and written in standard English?

Reviewer #3: Yes

Reviewer #4: Yes

6. Review Comments to the Author

Reviewer #3: The authors present a cross-sectional study investigating a novel and interesting research question. In fact, due to their widespread use in almost any society, studies addressing research questions related to mobile phones and the resulting RF-EMF exposure are of considerable interest for public health. In my view, the authors satisfactorily addressed the comments by Reviewer 1 in the first round of reviews, even though of course this primarily needs to be evaluated by Reviewer 1. Moreover, I agree with their reply to Reviewer 2; in my opinion, there is no doubt that an observational study should be classified as a research article. Nevertheless, I would recommend that the authors take into account the following questions and remarks before the manuscript is released for publication.

Abstract:

I would suggest to add the sampling strategy (convenience sampling) to the abstract. Ideally, apart from the number of participants also the response rate should be provided but I assume that this rate cannot be calculated because of the sampling method (making it vice versa even more important to mention the sampling strategy).

I would also suggest to add “by means of linear regression models” to the sentence “Data were analysed to examine if…”.

Could you, if the word count allows, provide specific numbers (regression coefficients, p-values) to the statement “Men who kept their handset away from the body parts […] were less likely to perceive health risks form their behaviour”?

Introduction:

Page 4, second sentence: Can you add the country where this study (women aged 15-40 years) was performed? I think this information may be of interest since in the previous sentence you mention cultural differences as to where individuals carry their phone.

Methods:

Page 5, first paragraph: As mentioned above, it would be interesting to have a response rate. Do you happen to have any information on how many individuals you approached? In any case, I think the strategy of using a convenience sampling should be mentioned as a limitation of the study in the discussion section.

Page 5, second paragraph: Could you specify “handset carrying location information” (for instance, for which specific locations did you ask?) or provide the full questionnaire in the supplement?

Results:

Page 9, second paragraph. The statement “If the mobile phone was not in use, 34.7% placed it in close contact with the body…” comes a little bit as a surprise when in the sentence before you state that when not in use, 54% placed it on a table/desk and 46% in the trouser pocket. Do the 34.7% refer to not in use outdoors? Can you perhaps clarify this?

In my view, the high proportion (78.7%) of participants owning a mobile for more than 20 years appears pretty high? As far as I am aware, even though mobile phones became available in the 1990s, their widespread use did not kick in before the early 2000s (although I am not sure how the situation is in Australia), indicating that the majority of your participants seem to be (very) early adopters. Do you think this may point towards some form of selection bias in the sense that you recruited a relatively tech-savvy sample (which may differ with respect to phone use or risk perception compared to the general population)? In my view, it would be worthwhile to elaborate on this issue in a few sentences in the discussion part.

What I also see as quite surprising is your finding that men carrying their phone close to the body reported higher risk perception levels than those placing it in a bag or table. Would you not expect that individuals more concerned about potential health effects carry their phone farther away from their body? I think this may also be an interesting question to be addressed in the discussion.

Discussion:

Page 13, first paragraph: “For example, up to 64% of men…” Can you add a reference for this study?

In general, I would recommend that in the first paragraph you highlight the most important findings of your study (including those on risk perception levels) and then start with the comparison to other studies in the second paragraph.

Page 15, first paragraph: I think the potential of selection bias is a more important aspect here then the question of the generalisability of your study to other parts of Australia. As mentioned above, the use of a convenience sample and the potential of selection bias, e.g. with respect to attraction to use of technical devices in general and mobile phones in particular, and potential implications for the interpretation of your numbers, should receive some attention. If I am not mistaken, there is some evidence from other studies on mobile phone use that individuals with higher levels of usage tend to have a higher interest to participate in such studies than non-regular users. On the other hand, it may also seem plausible that people with higher levels of concern regarding health effects are more prone to join such as study as yours (which may lead to an overestimation of the true level of risk perception in the general population). To make a long story short, it would be great if you could add your thoughts on those aspects to the discussion part.

Minor comments:

Abstract, second sentence: “… and to assess relationship…” should read “… and to assess its/the relationship…” (add “its” or “the”)

In the second part of the abstract, some percentages are provided with decimal places and some without. I would recommend to harmonise this.

Introduction, third sentence: “enabled” should read “enable” as all other sentences in this paragraph are framed in present tense.

Results, first sentence: The information that 356 men participated is redundant since the sample size is already mentioned in the methods section. I would thus recommend to remove the first sentence and add the age range to the sentence describing the mean (SD) age of participants.

Page 14, last sentence: Add a comma after “body”

Page 15, first paragraph: “eg winter” should read “e.g., winter”

Reviewer #4: I do think that the manuscript merits to be published on PlosOne. In my opinion, Authors correctly addessed the comments of Reviewer 1. Moreover, I agree with Authors on their answer to the comment of Reviewer 2.

I have only a minor point to raise:

Table 1, Education: what does it means TAFE? On the Internet you can find some clues, e.g. :Australian Technical and Further Education (TAFE) institutes are government-owned providers of VET courses (Vocational Education and Training). Authors have to be more explicit, also with the use of a footnote that permits to the readers to understand the meaning of that acronym (without surfing the Internet!)

7. PLOS authors have the option to publish the peer review history of their article (what does this mean?). If published, this will include your full peer review and any attached files.

Reviewer #3: **Yes: **Tobias Weinmann

Reviewer #4: **Yes: **Stefano Mattioli

---

## [Author Response · Author response to Decision Letter 1]

17 May 2022

Reviewer #3: 

The authors present a cross-sectional study investigating a novel and interesting research question. In fact, due to their widespread use in almost any society, studies addressing research questions related to mobile phones and the resulting RF-EMF exposure are of considerable interest for public health. In my view, the authors satisfactorily addressed the comments by Reviewer 1 in the first round of reviews, even though of course this primarily needs to be evaluated by Reviewer 1. Moreover, I agree with their reply to Reviewer 2; in my opinion, there is no doubt that an observational study should be classified as a research article. Nevertheless, I would recommend that the authors take into account the following questions and remarks before the manuscript is released for publication.

AR: Thank you

Abstract: I would suggest to add the sampling strategy (convenience sampling) to the abstract. Ideally, apart from the number of participants also the response rate should be provided but I assume that this rate cannot be calculated because of the sampling method (making it vice versa even more important to mention the sampling strategy).

I would also suggest to add “by means of linear regression models” to the sentence “Data were analysed to examine if…”.

Could you, if the word count allows, provide specific numbers (regression coefficients, p-values) to the statement “Men who kept their handset away from the body parts […] were less likely to perceive health risks form their behaviour”?

AR: We have added the analysis model in the abstract and it reads as, “Data were analysed using linear regression models to examine if risk perception differed by mobile phone carrying location”. In addition, we have included information about the levels of significance. We have also revised the limitation section of the discussion to reflect the sampling limitations (see the last paragraph of discussion section).

Introduction:

Page 4, second sentence: Can you add the country where this study (women aged 15-40 years) was performed? I think this information may be of interest since in the previous sentence you mention cultural differences as to where individuals carry their phone.

AR: We have now added information that the study was from Australia

Methods:

Page 5, first paragraph: As mentioned above, it would be interesting to have a response rate. Do you happen to have any information on how many individuals you approached? In any case, I think the strategy of using a convenience sampling should be mentioned as a limitation of the study in the discussion section.

AR: We have now added this to the limitation section and it reads as, “Participants were recruited via advertisements and may be pre-selected based on their access to the invitations and voluntary expression of their interest for participation and hence may not be representative of the source population.”

Page 5, second paragraph: Could you specify “handset carrying location information” (for instance, for which specific locations did you ask?) or provide the full questionnaire in the supplement?

AR: We have revised the paragraph and added the following information: “Mobile handset carrying locations of men were assessed by asking them to rate, on a Likert-scale (1 = never; 2 = rarely, 3 = occasionally, 4 = often, 5 = very often), how commonly they carry it with their hand, keep it on a table, in a bag, or in the pocket of their T-shirt, Jacket or Trousers (front or back).” 

Results:

Page 9, second paragraph. The statement “If the mobile phone was not in use, 34.7% placed it in close contact with the body…” comes a little bit as a surprise when in the sentence before you state that when not in use, 54% placed it on a table/desk and 46% in the trouser pocket. Do the 34.7% refer to not in use outdoors? Can you perhaps clarify this?

AR: We have revised the sentence and it now reads as, “Of the men who keep their mobile phone in close contact to their body parts, 46.0% keep it in the trouser front pocket.”

In my view, the high proportion (78.7%) of participants owning a mobile for more than 20 years appears pretty high? As far as I am aware, even though mobile phones became available in the 1990s, their widespread use did not kick in before the early 2000s (although I am not sure how the situation is in Australia), indicating that the majority of your participants seem to be (very) early adopters. Do you think this may point towards some form of selection bias in the sense that you recruited a relatively tech-savvy sample (which may differ with respect to phone use or risk perception compared to the general population)? In my view, it would be worthwhile to elaborate on this issue in a few sentences in the discussion part.

AR: Thank you for the comment. The high prevalence of use might be unique case to Australia. For example, in a study conducted over a decade ago 68% of children aged 10-11 years (in 2012), reported having owned or used a mobile phone.

What I also see as quite surprising is your finding that men carrying their phone close to the body reported higher risk perception levels than those placing it in a bag or table. Would you not expect that individuals more concerned about potential health effects carry their phone farther away from their body? I think this may also be an interesting question to be addressed in the discussion.

AR: We agree that one can expect risk may impact behaviour to some extent such as not to carry the handset close to reproductive organs if one perceived risk. However, being a cross-sectional study, it may be that one was prompted about the possible risks and rated a higher risk if already carrying the phone in close proximity to the body. This was discussed in the earlier version of the manuscript (discussion section: paragraph 2), and reads as, “There is some evidence that perception of risk may impact behaviour to some extent such as whether or not to carry the handset close to reproductive organs (8). Men who kept their mobile phone away from body-parts while it was not in use were half as likely to perceive health risks from the specific behaviour. If one carries his mobile phone close to body parts, he might be prompted by being inquired about possible negative effects and may rate higher levels of risk perception.”

Discussion:

Page 13, first paragraph: “For example, up to 64% of men…” Can you add a reference for this study?

AR: We have now cited a reference

In general, I would recommend that in the first paragraph you highlight the most important findings of your study (including those on risk perception levels) and then start with the comparison to other studies in the second paragraph.

AR: We have revised this and divided the paragraph accordingly. We have added the following to the discussion (parag 1), “Men who carried their handset in close contact with their body, perceived higher risks from RF-EMF exposure compared to those who kept it away from the body. However, they were less likely to perceive health risks from their mobile phone handset carrying behaviour.”

Page 15, first paragraph: I think the potential of selection bias is a more important aspect here then the question of the generalisability of your study to other parts of Australia. As mentioned above, the use of a convenience sample and the potential of selection bias, e.g. with respect to attraction to use of technical devices in general and mobile phones in particular, and potential implications for the interpretation of your numbers, should receive some attention. If I am not mistaken, there is some evidence from other studies on mobile phone use that individuals with higher levels of usage tend to have a higher interest to participate in such studies than non-regular users. On the other hand, it may also seem plausible that people with higher levels of concern regarding health effects are more prone to join such as study as yours (which may lead to an overestimation of the true level of risk perception in the general population). To make a long story short, it would be great if you could add your thoughts on those aspects to the discussion part.

AR: We have now added this to the limitation section and it reads as, “Participants were recruited via advertisements and may be pre-selected based on their access to the invitations and voluntary expression of their interest for participation and hence may not be representative of the source population.”

Minor comments:

Abstract, second sentence: “… and to assess relationship…” should read “… and to assess its/the relationship…” (add “its” or “the”)

AR: Thank you. We added “the”

In the second part of the abstract, some percentages are provided with decimal places and some without. I would recommend to harmonise this.

AR: Corrected

Introduction, third sentence: “enabled” should read “enable” as all other sentences in this paragraph are framed in present tense.

AR: Corrected

Results, first sentence: The information that 356 men participated is redundant since the sample size is already mentioned in the methods section. I would thus recommend to remove the first sentence and add the age range to the sentence describing the mean (SD) age of participants.

AR: We have now deleted the sentence describing the mean (SD) age of participants as it is already presented in the table. Sample sized also removed from results section.

Page 14, last sentence: Add a comma after “body”

AR: Corrected

Page 15, first paragraph: “eg winter” should read “e.g., winter”

AR: Corrected

Reviewer #4: 

I do think that the manuscript merits to be published on PlosOne. In my opinion, Authors correctly addessed the comments of Reviewer 1. Moreover, I agree with Authors on their answer to the comment of Reviewer 2.

I have only a minor point to raise:

Table 1, Education: what does it means TAFE? On the Internet you can find some clues, e.g. :Australian Technical and Further Education (TAFE) institutes are government-owned providers of VET courses (Vocational Education and Training). Authors have to be more explicit, also with the use of a footnote that permits to the readers to understand the meaning of that acronym (without surfing the Internet!)

AR: We have now replaced TAFE with Vocational training (diploma).

---

## [Decision Letter · Decision Letter 2]

23 May 2022

Mobile phone carrying locations and risk perception of men: a cross-sectional study

PONE-D-21-18441R2

Dear Dr. ZELEKE,

We’re pleased to inform you that your manuscript has been judged scientifically suitable for publication and will be formally accepted for publication once it meets all outstanding technical requirements.

Kind regards,

Prof. Anat Gesser-Edelsburg, Ph.D.

Academic Editor

PLOS ONE

Additional Editor Comments (optional):

Reviewers' comments:

Reviewer's Responses to Questions

**Comments to the Author**

1. If the authors have adequately addressed your comments raised in a previous round of review and you feel that this manuscript is now acceptable for publication, you may indicate that here to bypass the “Comments to the Author” section, enter your conflict of interest statement in the “Confidential to Editor” section, and submit your "Accept" recommendation.

Reviewer #3: All comments have been addressed

2. Is the manuscript technically sound, and do the data support the conclusions?

Reviewer #3: Yes

3. Has the statistical analysis been performed appropriately and rigorously? 

Reviewer #3: Yes

4. Have the authors made all data underlying the findings in their manuscript fully available?

Reviewer #3: Yes

5. Is the manuscript presented in an intelligible fashion and written in standard English?

Reviewer #3: Yes

6. Review Comments to the Author

Reviewer #3: The authors satisfactorily addressed all my comments and questions. I thus recommend the manuscript to be accepted for publication.

7. PLOS authors have the option to publish the peer review history of their article (what does this mean?). If published, this will include your full peer review and any attached files.

Reviewer #3: **Yes: **Tobias Weinmann

---

## [Editor Report · Acceptance letter]

30 May 2022

PONE-D-21-18441R2 

Mobile phone carrying locations and risk perception of men: a cross-sectional study 

Dear Dr. Zeleke:

I'm pleased to inform you that your manuscript has been deemed suitable for publication in PLOS ONE. Congratulations! Your manuscript is now with our production department. 

Kind regards, 

on behalf of

Prof. Anat Gesser-Edelsburg 

Academic Editor

PLOS ONE